# The Impact of Perioperative and Predisposing Risk Factors on the Development of Postoperative Delirium and a Possible Gender Difference

**DOI:** 10.3390/geriatrics7030065

**Published:** 2022-06-14

**Authors:** Maria Wittmann, Andrea Kirfel, Diane Jossen, Andreas Mayr, Jan Menzenbach

**Affiliations:** 1Department of Anesthesiology and Intensive Care Medicine, University Hospital Bonn, Venusberg-Campus 1, 53127 Bonn, Germany; andrea.kirfel@ukbonn.de (A.K.); diane.jossen@gmail.com (D.J.); jan.menzenbach@ukbonn.de (J.M.); 2Institute for Medical Biometry, Informatics and Epidemiology, University Hospital Bonn, Venusberg-Campus 1, 53127 Bonn, Germany; andreas.mayr@ukbonn.de

**Keywords:** postoperative delirium, risk factors, older adults, gender difference

## Abstract

(1) Background: Postoperative delirium (POD) is an undesirable event especially for older patients after surgery. Perioperative risks for POD development are multiple, but gender differences are still poorly considered. In this observational study, predisposing and precipitating risk factors of POD and the possible gender influence are distinguished. (2) Methods: This observational prospective trial enrolled 1097 patients in a tertiary hospital from September 2018 until October 2019. POD was considered positive, if one of the tests Confusion Assessment Method for ICU (CAM-ICU) or Confusion Assessment Method (CAM), 4 ‘A’s Test (4AT) or Delirium Observation Screening (DOS) scale was positive on one of five assessment days. (3) Results: POD incidence was 23.5% and the mean age of study population was 72.3 ± 7.3 years. The multiple logistic regression model showed a significant impact of age (Odds Ratio (OR) 1.74; 95% Confidence Interval (CI): 1.37–2.22), American Society of Anesthesiologists (ASA) (OR 1.67; 95% CI: 1.25–2.26), surgery risk (OR 2.10; 95% CI: 1.52–2.95) and surgery duration (OR 1.17; 95% CI: 1.07–1.28), ventilation time (OR 1.64; 95% CI: 1.27–2.24), as well as the male sex (OR 1.74; 95% CI: 1.37–2.22) on POD risk. (4) Conclusions: Perioperative and predisposing risk factors had an impact on the development of POD. The influence of male sex should be considered in future research.

## 1. Introduction

Postoperative delirium (POD) is an underdiagnosed and adverse event related to surgical procedures [1,2,3]. The incidence of delirium varies in a range from 11–51% [1,4]. This is related to the different surgical disciplines and the various study designs. POD is not only a postoperative complication itself, it also contributes to the development of other undesired complications. Often described in literature are a prolonged inpatient stay in intensive care and in hospital, long-term cognitive impairment and an increased risk of mortality [5,6,7,8,9]. Thus, POD places a major burden on patients themselves and on the limited resources of the health care system [10,11]. 

Literature shows a large number of perioperative risk factors for POD that can be predisposing as well as precipitating. In particular, age, cognitive impairment, comorbidity, sensorial deficits, malnutrition, polymedication, impaired functional status and frailty are described as predisposing risk factors. In addition, the risk and duration of surgery, ventilation time, and intensive care stay are risk markers for the development of POD. [4,12]. Also more frequently discussed is the influence of different surgical disciplines, with cardiac surgery often described as having a relatively high incidence of POD [13,14]. Less discussed were gender differences in relation to POD development. One study showed, based on adjusted regression results, that male sex is an important risk factor for delirium development during cardiac valve surgery [15]. 

It is non-controversial to say that POD plays an important role especially in the aging society and that perioperative management is important in terms of risk screening, preventive measures to be derived and standardized POD testing [12,16,17]. 

The results presented here were obtained from the study “PRe-Operative Prediction of postoperative DElirium by appropriate SCreening (PROPDESC)” [18,19]. This has developed a preoperatively applicable risk score in a prospective observational setting that is easy and quick to apply in clinical routine and also has a good predictive power. In order to take a closer look at the impact of perioperative risk factors on POD development, these were considered in more detail in the analysis presented here. The knowledge of these risk factors may help to identify patients who could profit from risk mitigation. Another objective of this analysis is to illustrate gender differences in relation to delirium development. 

## 2. Materials and Methods

### 2.1. Study Design and Participants

The PROPDESC trial was an observational prospective single center trial in a university hospital in Germany [19]. PROPDESC was mainly performed to develop and internally validate a predictive risk score for POD. The trial was conducted from September 2018 to October 2019 and enrolled 1097 patients of several surgery disciplines. PROPDESC was registered in the German Registry for Clinical Studies under the number DRKS00015715 and was approved by the local Ethics Committee at the Medical Faculty of the Rheinische Friedrich-Wilhelms-University of Bonn. Written informed content was obtained for each patient. The trial complied within the principles of the declaration of Helsinki.

Inclusion criteria contained patient´s age of 60 years or older and a planned surgery duration of minimum 60 min. Exclusion criteria were emergency procedures, language barriers or missing compliance with the study protocol.

### 2.2. Data Collection

Preoperative data collection was conducted in the anesthesia outpatient department and in the standard care wards. The following preoperative parameters were relevant for the analyses presented here: age, sex, body-mass-index (BMI), American Society of Anesthesiologists (ASA) Physical Status Classification System, Revised Cardiac Risk Index (rCRI), New York Heart Association Classification (NYHA), self-reported Metabolic Equivalent of Tasks (MET), surgical discipline and surgical risk. Surgical risk was transformed from a 5-level Johns-Hopkins classification to the 3-level modified Johns-Hopkins surgical classification [20,21]. Furthermore, surgery duration, ventilation time and the length of stay (LOS) in ICU were recorded.

Trained doctoral students carried out the postoperative evaluation in the ICU and standard care units. In order to collect the primary endpoint POD, an examination with different assessment instruments was conducted in the morning on 5 consecutive days after surgery, where appropriate after the end of sedation. Sedated patients with Richmond Agitation-Sedation Scale (RASS) < −3 were considered as not assessable and therefore the testing for POD was initiated after exceeding this level of sedation according to Confusion Assessment Method for ICU (CAM-ICU) [22,23]. Different assessments were carried out in parallel to detect POD. This methodology is based on the primary goal of the PROPDESC study to develop a predictive risk score and to avoid missing a positive endpoint. All patients were tested postoperatively with the Delirium Observation Screening (DOS) scale by questioning the nursing staff about the patient’s behavior in the last 24 hours in order not to miss any abnormalities which are to be considered as POD [24]. In addition to DOS, CAM-ICU was used in intensive care. In standard care ward, the 4 ‘A’s Test (4AT) and the CAM were used in addition to the DOS [25,26]. The positive endpoint POD was confirmed if one of the applied tests was positive on one of the 5 visit days. The definition of a completed POD assessment required a valid conduct of at least three of the five scheduled postoperative visits. Patients discharged before the third visit without diagnosed delirium were classified as non-delirious on the assumption that they would not develop delirium in their usual environment. Furthermore, patients who died before the end of the 5 visits and had not shown POD were removed from the group to be analyzed.

### 2.3. Statistical Analysis

The exploratory statistical analysis was performed using the statistical programming environment R. Continuous variables are presented with mean and standard deviation (sd±). Categorical variables are shown as numbers and percentages (%). Patients were divided into two groups (non-POD and POD) based on the POD endpoint. The differences between these groups regarding the characteristics were analyzed using the non-parametric Wilcoxon rank-sum test for continuous variables and the Fisher’s exact test was computed to check for independence for categorical variables.

First a univariate logistic regression was performed to analyze the unadjusted effect of gender as a risk factor for POD. Multiple logistic regression was performed afterwards to analyze the impact of perioperative risk factors, adjusting also the gender effect for potential confounders. POD entered the model as the binary outcome variable. The potential risk factors were incorporated as metric variables (ventilation time in days and surgery duration in hours) or as categorical ones (age in increments of 10, ASA classification, surgery risk and sex) and served as independent variables. To improve interpretability, (adjusted) odds ratios (OR) were generated via transformation from the regression coefficients and are reported with corresponding 95% confidence interval (CI).

## 3. Results

### 3.1. Participants

Of 1097 enrolled patients, 72 (6.6%) had no surgery and four (0.4%) have withdrawn the informed consent (Figure 1). Among these four patients were two who had released their data for score development, but not for further analysis in this area. Of the 1021 enrolled patients, 15 (1.4%) died within the postoperative visitation period without the positive endpoint POD and were removed from the dataset. 30 (2.7%) patients had less than three completed visits and no positive endpoint POD and were also excluded from the analysis. Thus, 976 patients were included in the analyses.

### 3.2. Characteristics Variables Related to POD

The mean age of the patient cohort was 72 (±7.3) years and the gender distribution was 375 (38%) women and 601 (62%) men. We divided these patients into two groups based on the presence or absence of tested delirium: the POD group (*n =* 229; 23.5%) and the non-POD group (*n =* 747; 76.5%). The characteristics of the enrolled patients in these two groups are presented in Table 1. Patients who developed POD were older (mean age 73 vs. 72 years; *p* = 0.010), had a higher ASA (level 3 and 4: 85% vs. 56%; *p* < 0.001), NYHA (level III and IV: 41% vs. 18%; *p* < 0.001), rCRI (level 3 and 4: 57% vs. 26%; *p* < 0.001) and a lower MET (level 1–4: 60% vs. 43%, level 5–10: 38% vs. 52%; *p* < 0.001). The preoperatively assessed surgical risk was also significantly higher in patients who developed delirium postoperatively (level 3: 68% vs. 35%; *p* <0.001). The highest incidence of POD was found in cardiac surgery patients at 60%. 

Beside the preoperative variables, the intra- and postoperative variables are also significantly different between the two groups. The surgery duration was 1.4 times longer in the POD patients (279 vs. 200 min.; *p* < 0.001). The ventilation time of the POD patients was 3.8 times longer than that of patients without POD development (31 vs. 8 h; *p* < 0.001). However, it should be mentioned here that outliers influence the mean ventilation time of POD patients. 

Of the 976 patients analyzed, 477 (49%) were postoperatively in ICU. With regard to the primary endpoint, the POD patients spent a mean of 148 h in ICU and the patients without POD only 22 h (*p* < 0.001). It should be added here that outliers in the POD group affect the mean ICU LOS. 

### 3.3. Gender Characteristics in Relation to POD

As already seen in the characteristics, there is a significant difference between women and men with regard to the incidence of delirium. Thus, characteristics were analyzed separately by gender with regard to POD development (Table 2). While the age difference between the POD and non-POD groups was significant in men (73 vs. 71 years; *p* = 0.032), it was not in women (75 vs. 73 years; *p* = 0.062). However, the scores obtained as surrogate parameters for morbidity showed significant differences in women as well as in men with respect to POD. In both gender groups, patients who developed POD were higher in ASA and NYHA classification, as well as in rCRI classification. Proportionally, more men (67%) were grouped in ASA levels 3 to 4 than women (56%) in this cohort. 39% of the men showed an rCRI index of 3–4 and only 24% of the women. As difference in the gender specific comparison, the percentage of higher rCRI among the delirious male patients is to be pointed out (rCRI 4: 29% vs. 14%).

Another contrast should be highlighted in the gender comparison. Male patients underwent high-risk surgery (classified risk 3) at a higher percentage than female patients (surgery risk 3: 47% vs. 36%). These results are comparable to the percentage distribution of surgical disciplines. It is necessary to mention that 33% of males underwent cardiac surgery and only 21% of females. Conversely, 46% of women underwent surgery in the orthopedic department and only 27% of men. In terms of POD development, surgical discipline and preoperatively classified surgical risk are also significant for each gender group (for each *p* < 0.001).

Furthermore, the gender-specific results showed a proportionally longer surgery and ventilation times in men, as well as a longer LOS in ICU. On average, POD-affected men underwent surgery 1.4 times longer (289 vs. 210 min; *p* < 0.001), were ventilated 4.4 times longer (35 vs. 8 h; *p* < 0.001), and stayed in the ICU 4.3 times longer (208 vs. 49 h; *p* < 0.001). Among women, these differences in POD development are also significant. They underwent surgery 1.4 times longer (251 vs.186 min; *p* < 0.001), ventilated 2.9 times longer (22 vs. 8 h; *p* < 0.001), and stayed 1.6 times longer in the ICU (111 vs. 70 h; *p* = 0.001) in association with a POD.

### 3.4. Impact of Several Risk Factors on the Development of POD

Logistic regression without further adjustment was performed to examine the influence of gender on POD development (Table 3). It showed an OR of 1.84 (95% CI: 1.34–2.55; *p* < 0.001) with the female reference variable.

The results of the multiple logistic regression (Table 4) with risk adjustment showed a significant impact on the development of POD for all included risk factors. The largest adjusted OR with 2.10 (95% CI: 1.52–2.95; *p* < 0.001) shows the surgery risk classification. Furthermore, the logit regression also indicated age in 10-year increments as a significant influencing factor on POD (adj. OR 1.74; 95% CI: 1.37–2.22; *p* < 0.001). The ASA classification as a surrogate marker for the morbidity of the patient clientele showed an adjusted OR of 1.67 (95% CI: 1.25–2.26; *p* = 0.001). The intra- and postoperative parameters of surgery duration (adj. OR 1.17; 95% CI: 1.07–1.28; *p* < 0.001) and ventilation time (adj. OR 1.64; 95% CI: 1.27–2.24; *p* = 0.001) also revealed a significant impact on the POD development.

## 4. Discussion

The results of the prospective observational study presented here showed an overall incidence in postoperative delirium of 23.5%. This is on average to the previously reported 11–51% [1]. The high proportion of cardiac surgery can explain the comparatively high incidence to other studies. About 21% of the women and 33% of the men underwent open cardiac surgery, which is known to be a significant risk factor for the development of POD [13,14,27]. Women after such a procedure showed a POD in 52% and the men in 64%. With regard to our results, however, it should be added that significantly more men underwent cardiac surgery and thus had a relatively higher incidence of delirium.

In terms of gender distribution, it appears that males (38%) in the total cohort were substantially more likely to have been delirious than females (21%). These findings are supported by the 2021 retrospective study of cardiac surgery patients by Wang et al. [15]. There, male gender was confirmed as a significant risk factor for delirium development. In addition, they also showed that male patients suffered from the hyperactive form of POD much more frequently than women. Another study in hip fracture patients by Oh et al. prospectively shows that male sex is a risk factor for POD development, even after risk adjustment [28]. Both studies noted and discussed this gender difference but were also unable to provide a conclusive explanation. In the two regression models, gender showed an OR of 1.84 in the univariate model and an adjusted OR of 1.59 in the multiple model with additional risk variables. This difference between the estimates is relatively small and suggests that male sex does represent some independent risk factor for POD development (as the effect does not vanish after adjustment), but this value is not greater than the other risk factors considered here. 

Besides gender, the variable of age also showed a significant difference in POD and non-POD patients (72 vs. 73 years, *p* = 0.010). Age has been reported more frequently in the literature as a risk factor for POD development and thus these results are not surprising. Likewise, comorbidities, which often increase with age, are considered as significant influencing factors. The surrogate parameters for comorbidity in the present study support this statement. ASA, NYHA, and rCRI levels differed significantly between the POD and non-POD groups (for each *p* < 0.001). 72% of delirious patients have ASA class 3 preoperatively and as many as 13% have ASA class 4. The European Society of Anesthesiology and Intensive Care (ESAIC) itself identifies ASA classification as a possible surrogate parameter regarding comorbidities in its POD guideline [12]. This suggests as conclusion that multimorbid patients are more prone to develop POD. Beyond that, ASA level ≥ 3 is also classified as a risk marker for POD development in other surgical disciplines [29,30]. Furthermore, logistic regression showed that an ASA level increase had an effect of 1.67 OR. Regarding the difference in incidence of delirium between the genders, it should be noted here that approximately 10% more men were classified in ASA 3 and 4 than women. Thus, the men showed a higher percentage of comorbidity in comparison to the women. This observation has also been made by Oh et al. in their delirium study [28]. In contrast to the comorbidities, however, it is striking here that in terms of percentage the men have a better functional capacity than women. With about 8% more, the men are classified in the level from 5 self-reported MET upwards. In a study of cardiorespiratory fitness and all-cause mortality, male participants also achieved comparatively higher METs than women, although the selected endpoint of survival was the same for both sexes [31]. This could suggest that self-perception of one’s fitness level might vary depending on gender.

Besides the predisposing POD factors, the precipitating factors also emerged as important markers in our study. Preoperatively graded surgical risk and actual duration of surgery differed significantly between POD and non-POD patients. The ESAIC POD Guideline recommended considering that the duration of surgery has a high influence on the development of POD [12]. Descriptive analysis of our study also showed that POD patients underwent surgery 1.4 times longer than patients without POD and also showed an OR of 1.17. A noticeable finding here was that male POD patients underwent surgery for an average of 40 min longer than female POD patients. The surgery risk classification showed the largest OR in the multiple regression model at 2.10. Of the total 420 patients admitted with surgical risk level 3, about 63% of them were cardiac surgery procedures alone. These results provide support for the statement that cardiac surgery and high-risk interventions influence the development of POD [12,13,27]. 

Based on the results of our study, it can be summarized that age and morbidity as predisposing factors have an influence on POD development. Furthermore, the precipitating factors such as surgery duration and surgery risk should be considered with regard to POD development. The influence of male sex as a risk factor was seen in the data presented here, but this issue needs further research. In a subgroup analysis of the PROPDESC study, we also showed that LOS in ICU has an impact on POD development [9].

There are limitations to this study that are briefly described below. The gold standard for diagnosing delirium would be an extensive examination by a psychiatrist, which however is usually not feasible in clinical routine of surgical patients. Therefore, POD assessment of this trial was also limited to the five-day assessment described above. Furthermore, our presented analysis is purely exploratory and all findings should be interpreted with care as the trial was not design to confirm any hypothesis regarding gender effects or any other risk factors. Another limitation to be mentioned is that although the regression analysis has included certain risk factors for postoperative delirium, there may be other unobserved confounders.

## 5. Conclusions

Predisposing (age and ASA classification) and precipitating perioperative (ventilation time, surgery risk and duration) risk factors had a significant impact on the development of POD. The influence of gender on risk of POD should be considered in future research. Standardized risk screening should be introduced into routine clinical practice, and risk prevention programs should be established for patients identified as being at higher risk for developing POD.

## Figures and Tables

**Figure 1 geriatrics-07-00065-f001:**
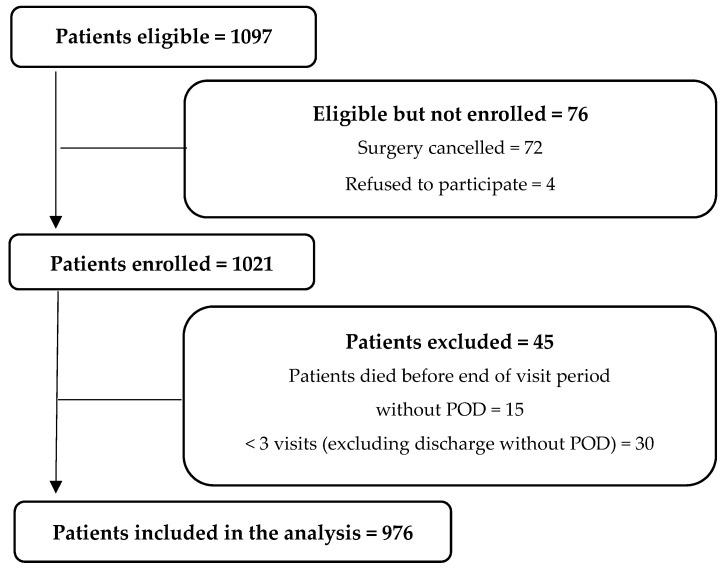
Flow diagram.

**Table 1 geriatrics-07-00065-t001:** Characteristics variables related to POD.

Characteristics	Total (*n* = 976)	Non-POD Group (*n* = 747)	POD-Group (*n* = 229)	*p* Value	Missing Values
Age (mean, sd)	72.3 ± 7.3	72.0 ± 7.3	73.3 ± 7.2	0.010	0
Sex				<0.001	0
female	375 (38.4)	311 (41.6)	64 (27.9)		
male	601 (61.6)	436 (58.4)	165 (72.1)		
BMI (mean, sd)	27.7 ± 5.4	27.8 ± 5.5	27.5 ± 5.0	0.827	3
ASA				<0.001	0
ASA 1	25 (2.6)	21 (2.8)	4 (1.7)		
ASA 2	339 (34.7)	308 (41.2)	31 (13.5)		
ASA 3	544 (55.7)	380 (50.9)	164 (71.6)		
ASA 4	68 (7.0)	38 (5.1)	30 (13.1)		
NYHA				<0.001	0
NYHA I	413 (42.3)	361 (48.3)	52 (22.7)		
NYHA II	336 (34.4)	252 (33.7)	84 (36.7)		
NYHA III	210 (21.5)	125 (16.7)	85 (37.1)		
NYHA IV	17 (1.7)	9 (1.2)	8 (3.5)		
rCRI				<0.001	0
rCRI 1	408 (41.8)	367 (49.1)	41 (17.9)		
rCRI 2	243 (24.9)	185 (24.8)	58 (25.3)		
rCRI 3	218 (22.3)	144 (19.3)	74 (32.3)		
rCRI 4	107 (11.0)	51 (6.8)	56 (24.5)		
MET				<0.001	0
MET < 1	11 (1.1)	9 (1.2)	2 (0.9)		
MET 1–4	457 (46.8)	320 (42.8)	137 (59.8)		
MET 5–10	475 (48.7)	389 (52.1)	86 (37.6)		
MET > 10	33 (3.4)	29 (3.9)	4 (1.7)		
Surgical discipline				<0.001	0
Others	193 (19.8)	174 (23.3)	19 (8.3)		
Cardiac Surgery	274 (28.1)	136 (18.2)	138 (60.3)		
Orthopedic Surgery	337 (34.5)	294 (39.4)	43 (18.8)		
Thoracic Surgery	21 (2.2)	17 (2.3)	4 (1.7)		
Abdominal Surgery	123 (12.6)	107 (14.3)	16 (7.0)		
Vascular Surgery	28 (2.9)	19 (2.5)	9 (3.9)		
Surgical risk				<0.001	0
low	126 (12.9)	123 (16.5)	3 (1.3)		
Intermediate	430 (44.1)	360 (48.2)	70 (30.6)		
high	420 (43.0)	264 (35.3)	156 (68.1)		
Surgery duration (min) (mean, sd)	218.4 ± 125.4	200.0 ± 120.4	278.5 ± 122.8	<0.001	1
Ventilation time (h) (mean, sd)	13.3 ± 56.2	7.8 ± 12.6	31.4 ± 111.7	<0.001	11
LOS ICU (h) (mean, sd)	51.3 ± 226.0	22.0 ± 61.5	147.6 ± 442.1	<0.001	10

Data are number (%) unless stated otherwise. POD = Postoperative delirium, BMI = body mass index, ASA = American Society of Anesthesiology, NYHA = New York Heart Association, rCRI = Revised Cardiac Risk Index, MET = Metabolic Equivalent of Tasks, LOS = length of stay, ICU = Intensive Care Unit.

**Table 2 geriatrics-07-00065-t002:** Gender specific characteristics in relation to POD.

	Women (*n* = 375)			Men (*n* = 601)		
Characteristics	Non-POD Group (*n* = 311)	POD-Group (*n* = 64)	*p* Value	Non-POD Group (*n* = 436)	POD-Group (*n* = 165)	*p* Value
Age (mean, sd)	72.8 ± 7.7	74.6 ± 6.3	0.062	71.4 ± 7.0	72.8 ± 7.5	0.032
BMI (mean, sd)	27.7 ± 6.3	26.8 ± 5.7	0.532	27.9 ± 4.9	27.7 ± 4.6	0.843
ASA			<0.001			<0.001
ASA 1	9 (2.9)	0 (0.0)		12 (2.8)	4 (2.4)	
ASA 2	144 (46.3)	11 (17.2)		164 (37.6)	20 (12.1)	
ASA 3	144 (46.3)	44 (68.8)		236 (54.1)	120 (72.7)	
ASA 4	14 (4.5)	9 (14.1)		24 (5.5)	21 (12.7)	
NYHA			0.001			<0.001
NYHA I	142 (45.7)	19 (29.7)		219 (50.2)	33 (20.0)	
NYHA II	109 (35.0)	18 (28.1)		143 (32.8)	66 (40.0)	
NYHA III	58 (18.6)	24 (37.5)		67 (15.4)	61 (37.0)	
NYHA IV	2 (0.6)	3 (4.7)		7 (1.6)	5 (3.0)	
rCRI			<0.001			<0.001
rCRI 1	169 (54.3)	20 (31.3)		198 (45.4)	21 (12.7)	
rCRI 2	83 (26.7)	14 (21.9)		102 (23.4)	44 (26.7)	
rCRI 3	45 (14.5)	21 (32.8)		99 (22.7)	53 (32.1)	
rCRI 4	14 (4.5)	9 (14.1)		37 (8.5)	47 (28.5)	
MET			0.045			0.001
MET < 1	5 (1.5)	0 (0.0)		4 (0.9)	2 (1.2)	
MET 1–4	149 (47.9)	43 (67.2)		171 (39.2)	94 (57.0)	
MET 5–10	146 (46.9)	20 (31.3)		243 (55.7)	66 (40.0)	
MET > 10	11 (3.5)	1 (1.6)		18 (4.1)	3 (1.8)	
Surgical discipline			<0.001			<0.001
Others	44 (14.1)	8 (12.5)		130 (29.8)	11 (6.7)	
Cardiac Surgery	44 (14.1)	33 (51.6)		92 (21.1)	105 (63.6)	
Orthopedic Surgery	157 (50.5)	17 (26.6)		137 (31.4)	26 (15.8)	
Thoracic Surgery	8 (2.6)	1 (1.6)		9 (2.1)	3 (1.8)	
Abdominal Surgery	52 (16.7)	2 (3.1)		55 (12.6)	14 (8.5)	
Vascular Surgery	6 (1.9)	3 (4.7)		13 (3.0)	6 (3.6)	
Surgical risk			<0.001			<0.001
low	51 (16.4)	1 (1.6)		72 (16.5)	2 (1.2)	
Intermediate	161 (51.8)	27 (42.2)		199 (45.6)	43 (26.1)	
high	99 (31.8)	36 (56.3)		165 (37.8)	120 (72.7)	
Surgery duration (min) (mean, sd)	185.5 ± 118.1	250.5 ± 122.1	<0.001	210.3 ± 121.1	289.4 ± 121.7	<0.001
Ventilation time (h) (mean, sd)	7.6 ± 15.4	22.0 ± 54.3	<0.001	7.9 ± 10.1	35.0 ± 127.2	<0.001
LOS ICU (h) (mean, sd)	69.8 ± 118.0	111.4 ± 124.6	0.001	48.9 ± 64.1	208.3 ± 552.9	<0.001

Data are number (%) unless stated otherwise. POD = Postoperative delirium, BMI = body mass index, ASA = American Society of Anesthesiology, NYHA = New York Heart Association, rCRI = Revised Cardiac Risk Index, MET = Metabolic Equivalent of Tasks, LOS = length of stay, ICU = Intensive Care Unit.

**Table 3 geriatrics-07-00065-t003:** Univariate logistic regression analysis with sex (POD vs. non-POD group).

	OR	95% CI		*p*-Value
Sex (ref. women)	1.84	1.34	2.55	<0.001

OR—Odds Ratio, CI—Confidence Interval.

**Table 4 geriatrics-07-00065-t004:** Multiple logistic regression analysis (POD vs. non-POD group).

	Adj. OR	95% CI		*p*-Value
Surgery duration (h)	1.17	1.07	1.28	<0.001
Ventilation time (day)	1.64	1.27	2.24	0.001
Surgery risk	2.10	1.52	2.95	<0.001
Age (10 years)	1.74	1.37	2.22	<0.001
ASA	1.67	1.25	2.26	0.001
Sex (ref. women)	1.59	1.11	2.28	0.012

adj. OR—adjusted Odds Ratio, CI—Confidence Interval, ASA—American Society of Anesthesiology classification; 12 values deleted due to missings.

## Data Availability

The data sets generated and analysed during the study are available from the corresponding author on reasonable request. The R code used for the analysis is available from the corresponding author on reasonable request.

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
