# Peer review of "The Impact of Perioperative and Predisposing Risk Factors on the Development of Postoperative Delirium and a Possible Gender Difference"

_geriatrics, 2022, doi:10.3390/geriatrics7030065_

Round 1

Reviewer 1 Report

Reviewer Comments:

Thank you for your well-structured and important work. Postoperative delirium is not a new topic in geriatric medicine, but a prospectively designed study covering 1097 postoperative patients is definitely impressive. 

Abstract:

Background: “POD is an adverse event for older adults…”

Isn´t POD an undesirable (adverse) event in all (not only older) adults? Please correct: either: “undesirable event especially for older (or geriatric) patients” or keep in in general: “..adverse event after surgery”.

“The end point tested POD was considered positive…. “. I would suggest to delete “the end point tested”.

“The multiple logistic regression model showed a significant impact on POD from…” Shouldn´t it be “… significant impact of age…. on POD”.

Main body:

There are a few language misunderstandings: in hospital (better in-hospital), longer-term (better longterm), uncontroversial (better non-controversial), p3 l120: has withdrawn (should be have withdrawn), delirium incidence (better incidence of delirium), in percentage terms (better in terms of percentage)…. that I encourage the authors to ask a native speaker to correct for vocabulary errors.

In general:

I don´t really understand, why gender is referred to as a possible risk factor, although a multiple logistic regression analysis has proven a significant finding. Furthermore, wouldn´t it be necessary to include type of surgery, MET, cCRI, NYHA, and LOS ICU (see table 2)  within your multiple regression analysis, since you stated that male patients showed significantly different results compared to female patients?

Further ideas for improvement are described below.

Data Collection 2.2

You write that you recruited patients in the ward and in the outpatient clinic. Is there an average value of how many patients have already been hospitalized preoperatively for how many days?

Would it be possible to distinguish the presence of hypo- or hyperactive POD by collecting the RASS (CAM ICU)?

Results 3.2, Lines 180 ff.

You write here that the scores (surrogate parameters for morbidity) differ significantly. Could you describe this in more detail in the text.

Results 3.3

Table 4 shows the multiple logistic regression with adjusted risk factors. It would be easier for the reader to understand if you would also refer to the adjustments taken care for an describe these in der body text of the results or in the methods section.

Discussion, Line 225

“..presented here showed a discipline-independent delirium incidence of…” This is misleading since you describe differences between cardia surgery and for example thoracic surgery. I guess you wanted to say “..showed an overall incidence in postoperative delirium of 23.5%.”

Discussion, Line 238

“Another prospective study…” Since you write about a retrospective study previously, this wording is again misleading. Maybe: “A further study investigating hip fracture patients via a prospective design…”

Discussion, Line 254

You refer to the ESAIC guideline. It would be better for the reader to introduce the abbreviation first.

Discussion, Line 284

It would be better to stay with the previous wording throughout the paper. Previously “surgery duration” and here “operation time”.

Conclusion

The last sentence is horrible.

Maybe: “Male sex revealed to correlate with higher incidences in POD and should therefore be taken into account when stratifying risk scores in future investigations.

Best regards

Author Response

Reviewer 1 Comments:

Thank you for your well-structured and important work. Postoperative delirium is not a new topic in geriatric medicine, but a prospectively designed study covering 1097 postoperative patients is definitely impressive. 

Dear Reviewer, thank you for this valuation.

Abstract:

Background: “POD is an adverse event for older adults…”

Isn´t POD an undesirable (adverse) event in all (not only older) adults? Please correct: either: “undesirable event especially for older (or geriatric) patients” or keep in in general: “..adverse event after surgery”.

In lines 12 to 13, we changed to:

“undesirable event especially for older (or geriatric) patients”

“The end point tested POD was considered positive…. “. I would suggest to delete “the end point tested”.

In line 17, we deleted “the end point tested”.

“The multiple logistic regression model showed a significant impact on POD from…” Shouldn´t it be “… significant impact of age…. on POD”.

In lines 21 and 24 we changed to “… significant impact of age…. on POD”.

Main body:

There are a few language misunderstandings: in hospital (better in-hospital), longer-term (better longterm), uncontroversial (better non-controversial), p3 l120: has withdrawn (should be have withdrawn), delirium incidence (better incidence of delirium), in percentage terms (better in terms of percentage)…. that I encourage the authors to ask a native speaker to correct for vocabulary errors.

In line 35, we changed to in-hospital and long-term.

In line 49, we changed to non-controversial.

In line 126, we changed to have withdrawn.

In lines 183, 237 and 272 we changed to incidence of delirium.

In line 277, we changed to in terms of percentage.

In general:

I don´t really understand, why gender is referred to as a possible risk factor, although a multiple logistic regression analysis has proven a significant finding. Furthermore, wouldn´t it be necessary to include type of surgery, MET, cCRI, NYHA, and LOS ICU (see table 2)  within your multiple regression analysis, since you stated that male patients showed significantly different results compared to female patients?

Thank you very much for the assessment of our results, The reviewer is correct, the term “possible risk factor” might be partly misleading given our results. Also regarding the adjustement via the multiple logistic regression model for other potential confounders (type of surgery, MET, cCRI, NYHA, LOS ICU) the reviewer is absolutely right: also these other factors might be relevant for adjustment. Actually we tried different sets of variables to include in our model, leading to overall similar results. The reason we can not just include all potential confounders is twofold: First, although we have a relatively large sample size – the number of cases is the main figure describing the degrees of freedom for a logistic regression model. Given the POD incidence in our trial (n =229) it is numerically not possible to adjust for all these (partly multi-categorical) variables at the same time. Second, these different risk classifications and particularly also the LOS variables are highly correlated among each others – leading to multicollinearity issues when included together in the model (which results in further numerical instabilities).     

In line 316 we deleted the word “possible”

 Further ideas for improvement are described below.

Data Collection 2.2

You write that you recruited patients in the ward and in the outpatient clinic. Is there an average value of how many patients have already been hospitalized preoperatively for how many days?

Thank you for this question. Patients were enrolled in the PROPDESC study between 60 and 1 day preoperatively. Unfortunately, the data do not provide a specific distribution between the prehospital and inpatient included patients.

Would it be possible to distinguish the presence of hypo- or hyperactive POD by collecting the RASS (CAM ICU)?

We have already discussed this interesting question in advance in the team. The RASS was collected from our patients in the ICU. However, many patients in the ICU also received sedatives, so we would not consider the RASS to be valid enough to determine hypo- or hyperactivity.

Results 3.2, Lines 180 ff.

You write here that the scores (surrogate parameters for morbidity) differ significantly. Could you describe this in more detail in the text.

We added the following in the lines 188 – 191:

“In both gender groups, patients who developed POD were higher in ASA and NYHA classification, as well as in rCRI classification. Proportionally, more men (67%) were grouped in ASA levels 3 to 4 than women (56%) in this cohort. 39% of the men showed an rCRI index of 3 - 4 and only 24% of the women.”

Results 3.3

Table 4 shows the multiple logistic regression with adjusted risk factors. It would be easier for the reader to understand if you would also refer to the adjustments taken care for an describe these in der body text of the results or in the methods section.

 We added the following in this result part in line 220:

“… with risk adjustment…”

In the results section, we have added "adjusted" in front of each OR.

Discussion, Line 225

“..presented here showed a discipline-independent delirium incidence of…” This is misleading since you describe differences between cardia surgery and for example thoracic surgery. I guess you wanted to say “..showed an overall incidence in postoperative delirium of 23.5%.”

In lines 236 – 237 we changed to “..showed an overall incidence in postoperative delirium of 23.5%.”

Discussion, Line 238

“Another prospective study…” Since you write about a retrospective study previously, this wording is again misleading. Maybe: “A further study investigating hip fracture patients via a prospective design…”

In lines 250 - 251, we changed the following sentence:

“Another study in hip fracture patients by Oh et al. prospectively shows that male sex is a risk factor for POD development, even after risk adjustment”

Discussion, Line 254

You refer to the ESAIC guideline. It would be better for the reader to introduce the abbreviation first.

In lines 266 – 267 we added the whole organisation name “European Society of Anaesthesiology and Intensive Care”

Discussion, Line 284

It would be better to stay with the previous wording throughout the paper. Previously “surgery duration” and here “operation time”.

We changed the “operation” to “surgery” in line 297.

 Conclusion

The last sentence is horrible.

Maybe: “Male sex revealed to correlate with higher incidences in POD and should therefore be taken into account when stratifying risk scores in future investigations.

In line 311 we changed the sentence to:

The influence of gender on risk of POD should be considered in future research.

Reviewer 2 Report

The authors studied the risk factors of POD using the data collected for their published study (“PRe-Operative Prediction of postoperative DElirium by appropriate SCreening (PROPDESC)"). They concluded that age, ASA classification, ventilation time, surgery risk and duration, and sex showed a significant impact on the development of POD based on multiple logistic regression analysis. Their previous study (PROPDESC) was performed to develop a pragmatic risk screening score for POD and reported high prediction accuracy by selecting a dozen of risk factors, including age, sex, BMI, ASA Classification, Revised Cardiac Risk Index, New York Heart Association Classification (NYHA), Metabolic Equivalent of Tasks (MET), surgical risk, surgical discipline, long-term medication, and preoperative laboratory values. In the article, they concluded that the PROPDESC score showed promising performance on a separate validation cohort in predicting POD based on routine preoperative data.

Major points:

-To re-analyze the data that were collected for the original questions or hypotheses, justification must be provided both scientifically and statistically. And the results should be discussed with respect to the original study and its results. This aspect may need to be discussed in the limitation as well.

-Overall, the statistical analysis needs to be explained in more details for the methods as well as for the results, including for the tables. It is not clear why the authors used logistic regression only for gender, but “multiple regression analysis” for all factors, which requires justification. Gender should be analyzed as one of the factors in the multiple regression or the other appropriate analytical method for multiple factors. If the authors intend to study the effect on gender on POD, it may be necessary to perform a separate clinical study with an appropriate design and power to determine the gender effect. I would strongly recommend involving a statistician for this manuscript if this may be revised.

Author Response

The authors studied the risk factors of POD using the data collected for their published study (“PRe-Operative Prediction of postoperative DElirium by appropriate SCreening (PROPDESC)"). They concluded that age, ASA classification, ventilation time, surgery risk and duration, and sex showed a significant impact on the development of POD based on multiple logistic regression analysis. Their previous study (PROPDESC) was performed to develop a pragmatic risk screening score for POD and reported high prediction accuracy by selecting a dozen of risk factors, including age, sex, BMI, ASA Classification, Revised Cardiac Risk Index, New York Heart Association Classification (NYHA), Metabolic Equivalent of Tasks (MET), surgical risk, surgical discipline, long-term medication, and preoperative laboratory values. In the article, they concluded that the PROPDESC score showed promising performance on a separate validation cohort in predicting POD based on routine preoperative data.

Dear reviewer, thank you for this helpful comments. Please find below our responses.

Major points:

-To re-analyze the data that were collected for the original questions or hypotheses, justification must be provided both scientifically and statistically. And the results should be discussed with respect to the original study and its results. This aspect may need to be discussed in the limitation as well.

Thank you for highlighting this important aspect of our trial! The reviewer is correct, we performed the analysis on data that was originally collected to develop a new risk score for POD. The trial design, the initial separation in training and validation cohort and also the sample size was not motivated by a trial hypothesis but by the development and evaluation of this risk score. In the primary analysis, we therefore did perform prediction modelling  – and no hypothesis testing w.r.t. risk factors was carried out. Nevertheless, the reviewer is of course right: Our presented analysis is purely exploratory and all findings should be interpreted with care as the trial was not design to confirm any hypothesis regarding gender effects or any other risk factors. We added this to the limitations paragraph in the Discussion of the revised version of the manuscript.        Menzenbach, J.; Guttenthaler, V.; Kirfel, A.; Ricchiuto, A.; Neumann, C.; Adler, L.; Kieback, M.; Velten, L.; Fimmers, R.; Mayr, A.; et al. Estimating Patients’ Risk for Postoperative Delirium from Preoperative Routine Data - Trial Design of the PRe-Operative Prediction of Postoperative DElirium by Appropriate SCreening (PROPDESC) Study - A Monocentre Prospective Observational Trial. Contemporary Clinical Trials Communications 2020, 17, 100501, doi:10.1016/j.conctc.2019.100501

-Overall, the statistical analysis needs to be explained in more details for the methods as well as for the results, including for the tables. It is not clear why the authors used logistic regression only for gender, but “multiple regression analysis” for all factors, which requires justification. Gender should be analyzed as one of the factors in the multiple regression or the other appropriate analytical method for multiple factors. If the authors intend to study the effect on gender on POD, it may be necessary to perform a separate clinical study with an appropriate design and power to determine the gender effect.

Thank you for the careful checking of the analysis. The motivation to perform first a simple logistic regression analysis with gender as only predictor and afterwards a multiple logistic regression was particularly to adjust gender effect for potential confounders. In our view, our results also provide evidence that parts of the strong gender differences in POD risk can be explained actually by different surgical risks of the two groups. However, parts of the gender effect still remain (the adjusted OR is smaller but still not equal to one).  The reviewer is of course correct, the analysis is still exploratory and results should not interpreted in a  confirmatory fashion.

In the revised version of the manuscript, we now motivate clearer the reasoning for this two-step analysis of the gender effect. So we added and adapted in the methods in lines 116 to 118 the following sentences: “First a univariate logistic regression was performed to analyse the unadjusted effect of gender as a risk factor for POD. Multiple logistic regression was performed afterwards to analyse the impact of perioperative risk factors, adjusting also the gender effect for potential confounders.”

With our study results we want to draw attention to a possible gender impact on the risk of POD. We agree, as we wrote in our conclusion, that this possible effect should be proven in further trials.

I would strongly recommend involving a statistician for this manuscript if this may be revised.

The author "Prof. Dr. Andreas Mayr"  who was responsible for the study design, the calculation of the sample size and supervised the various multiple analyses is a professor for Epidemiology and a statistician by training.

Reviewer 3 Report

Thanks for recommending me as a reviewer. Postoperative delirium (POD) is an adverse event for older adults after surgery. Perioperative risks for POD development are multiple, but gender differences are still poorly considered. In this observational study, predisposing and precipitating risk factors of POD and the possible gender influence are distinguished. If authors complete minor revisions, the quality of the study will be further improved.

  1. If the authors describe in more detail the trends in prior research related to predisposing risk factors on the development of postoperative delirium in the introduction section, it may help readers to understand.
  2.  Authors should more clearly state the purpose of their research in the introduction section.
  3. If the authors were more specific in the conclusion section, it could help readers understand.

Author Response

Thanks for recommending me as a reviewer. Postoperative delirium (POD) is an adverse event for older adults after surgery. Perioperative risks for POD development are multiple, but gender differences are still poorly considered. In this observational study, predisposing and precipitating risk factors of POD and the possible gender influence are distinguished. If authors complete minor revisions, the quality of the study will be further improved.

 Dear Reviewer, thank you for the nice feedback.

  1. If the authors describe in more detail the trends in prior research related to predisposing risk factors on the development of postoperative delirium in the introduction section, it may help readers to understand.

In the lines 39 to 41, we adapted the following sentence to:

“In particular, age, cognitive impairment comorbidity, sensorial deficits, malnutrition, polymedication, impaired functional status and frailty are described as predisposing risk factors”

  1.  Authors should more clearly state the purpose of their research in the introduction section.

In the introduction part, we added the following sentence in lines 57 – 58:

The knowledge of these risk factors may help to identify patients who could profit from risk mitigation.

  1. If the authors were more specific in the conclusion section, it could help readers understand.

In the conclusion part, we added the following sentence in lines 312 – 314:

“Standardized risk screening should be introduced into routine clinical practice, and risk prevention programs should be established for patients identified as being at higher risk for developing POD.”